# What Are the Neural Correlates of Impaired Awareness of Social Cognition and Function in Dementia? A Systematic Review

**DOI:** 10.3390/brainsci12091136

**Published:** 2022-08-26

**Authors:** Anna Hengstschläger, Andrew Sommerlad, Jonathan Huntley

**Affiliations:** 1Division of Psychiatry, University College London, London W1T 7BN, UK; 2Camden and Islington NHS Foundation Trust, London NW1 0PE, UK

**Keywords:** dementia, social cognition, social function, awareness, neuroimaging

## Abstract

Deficits in social cognition and function are characteristic of dementia, commonly accompanied by a loss of awareness of the presence or extent of these deficits. This lack of awareness can impair social relationships, increase patients’ and carers’ burden, and contribute to increased rates of institutionalization. Despite clinical importance, neural correlates of this complex phenomenon remain unclear. We conducted a systematic search of five electronic databases to identify functional and structural neuroimaging studies investigating the neural correlates of impaired awareness of social cognition and function in any dementia type. We rated study quality and conducted a narrative synthesis of the results of the eight studies that met the predefined eligibility criteria. Across these studies, deficits in awareness of impairments in social cognition and function were associated with structural or functional abnormalities in the frontal pole, orbitofrontal cortex, temporal pole, middle temporal gyrus, inferior temporal gyrus, fusiform gyrus, amygdala, hippocampus, parahippocampal gyrus, and insula. Several identified regions overlap with established neural correlates of social cognition. More research is needed to understand awareness of social cognition and function and how this becomes impaired in dementia to improve neuroscientific understanding, aid the identification of this problematic symptom, and target interventions to reduce burden and improve care.

## 1. Introduction

The focus of this systematic review is to examine the neural correlates of impaired awareness of deficits in social cognition and function in people with dementia. Deficits in social cognition and function are common features in neurodegenerative diseases and may worsen with disease progression [1,2,3]. The Diagnostic and Statistical Manual of Mental Disorders (DSM-5) criteria for dementia cites social cognition as one of the core domains of impaired neurocognitive function [4] and, in frontotemporal dementia (FTD), early decline in social conduct is a core diagnostic feature [5], with changes in social cognitive abilities commonly reported in behavioural-variant FTD (bvFTD) and semantic dementia (SD) [6,7]. Impairments of social cognitive processes and social function are also common in Alzheimer’s disease (AD), though are usually not as severe and occur later in the disease course than in FTD [2,8,9,10] and social cognitive decline also occurs in dementia with Lewy bodies (DLB) [11].

Social cognition refers to the cognitive processes necessary to perceive, process, and store social information from others, as well as to understand oneself in relation to others, and is critical for successful social interactions [12]. It can be conceptualized as comprising Theory of Mind (ToM), empathy, and emotion recognition [13]. ToM involves understanding one´s own and other people´s cognitive and emotional mental states (e.g., their desires, intentions, or beliefs), and knowing that these can differ from each other [13,14,15]. Empathy relates to a person’s ability to share the thoughts and feelings experienced by another person [15,16,17]. Emotion recognition refers to the process of recognizing emotions in other people [3]. These social cognitive processes work together and are key components of successful social function [18]. Social function relates to ‘how individuals associate and interact, both in society at large and in their own personal environment’ [19] and social functional interactions occur in many areas of everyday life [18,20,21]. Changes and deficits in social cognitive processes such as impairments in emotion recognition or ToM may lead to abnormal or disinhibited, socially inappropriate behaviour [10] and other changes in social function.

People with dementia commonly show impaired awareness of their illness and a lack of insight into the deterioration in their cognition and behaviour in a range of domains such as memory, function, personality change, and social cognitive processes [22,23,24,25]. Awareness refers to the ability to appropriately perceive and appraise a given aspect of one´s own specific situation, function, or performance [24,26]. Rather than being a unified phenomenon, impaired awareness in dementia is complex and multidimensional and people with dementia may lack awareness in some areas of function, but retain accurate awareness of their abilities in other domains [27,28]. Lack of awareness of cognitive or functional deficits is problematic for patients with dementia as it may compound the loss of function by preventing a person from seeking help from family, friends, or healthcare professionals for an impairment, or lead to them resisting offers of help. In addition, loss of awareness of deficits in dementia leads to increased caregiver burden, depression, and potentially burnout [24,25,29,30], leading in turn to higher rates of institutionalization and greater care costs [31,32].

The focus of this review is to examine the underlying neural correlates of impaired awareness of changes in social cognition and function in dementia. Impaired awareness of deficits in social cognition and function may be particularly problematic as it may lead to inappropriate behaviour, be distressing for families, and places extra burden on family members in social settings. Whilst there are several different questionnaires to measure awareness of memory function, relatively few validated tools are available to assess awareness of social cognition or social function [10,21,33]. The clinical measurement of deficits in awareness of social cognition and function are commonly based on: (a) clinician or informant ratings, in which the clinician judges the patient’s level of awareness of social cognitive processes; (b) self-appraisal performance discrepancy, in which a person with dementia gives either a prospective or retrospective judgment about performance on an assessment of social cognition and this prediction is compared with a person’s objective score; or (c) patient–informant discrepancy scores, where assessments of social cognition and function are completed by both the patient and an informant and the discrepancy between the scores is used as a measure of awareness, with an assumption that a non-cognitively impaired informant will give an accurate appraisal [23]. These tools aim to identify impairments in awareness of social cognition and function in people with dementia rather than the underlying impairments in social cognition per se. From a neurocognitive and neurobiological perspective, it remains unclear whether higher level awareness of deficits in social cognition and function can be dissociated from underlying processes of social cognition. For ToM, for example, awareness is assumed to be involved in the process of understanding that another person’s perspectives may not be the same as one’s own, and empathy shares similar underlying neural correlates with awareness of empathic abilities [12]. Given the dual burden of impaired social cognitive function and the lack of awareness of these deficits, it is important to clarify how these processes overlap [34].

Although a lack of awareness of social cognition and function deficits is characteristic of dementia, the underlying neural correlates are unclear. Understanding these neural underpinnings may contribute to understanding how and when neuropathological changes map onto clinical symptoms, as well as helping to clarify to what extent awareness is an essential aspect of social cognition, as some have proposed [4,13].

Social cognition is associated with activity in multiple brain regions, including the orbitofrontal cortex (OFC), medial prefrontal cortex, temporo-parietal junction, posterior and anterior cingulate cortex, medial parietal cortex, superior temporal sulcus, and the insula and anterior temporal cortex [9,18,35,36]. Different domains of social cognition may have specific neural substrates, for example, empathy is associated with the insula, whereas emotion recognition is correlated with the right somatosensory cortex [36], and ToM dysfunction is associated with temporoparietal junction impairment [10]. In lesion studies, damages to the OFC have been reported to be associated with changes in emotional and social behaviour related to impairments in emotional response and processing of facial or vocal expressions [37]. There is some overlap in these domains with neural regions associated with an awareness of illness or cognitive impairment in Alzheimer’s disease [38]. As stated above, awareness is not a unitary concept and neural correlates of awareness may differ according to the domain or object of awareness [33,39]. This is supported by a review of neural correlates of awareness in FTD which found that rather than awareness being a general or unified phenomenon, different objects or domains of awareness (e.g., memory, presence of disease, and social cognition) were associated with functional and structural changes in different brain regions [26]. However, it is unclear if deficits in awareness of social cognition and function in other subtypes of dementia are similarly associated with domain-specific neural correlates.

As these deficits have clinical relevance, the identification of possible neural correlates may help to adapt the care and management of patients [32]. In particular, knowledge about the biological basis of changes in social conduct may improve psychoeducation for relatives and caregivers and may increase their level of understanding.

In this systematic review, we therefore aimed to examine all existing structural and functional neuroimaging studies to identify neural correlates associated with an impaired awareness of social cognition and function in dementia.

## 2. Methods

### 2.1. Search Strategy

We performed an electronic literature search across five databases: MEDLINE, MEDLINE In-process, PsychINFO, Embase and Web of Science, from inception until 29th May 2020. No search limits were applied and no restrictions regarding year of publication were made. The search strategy was based on PICOS (population, intervention, control, outcome and study design) terms and key concepts related to the research question. The Boolean operators “AND” and “OR” were used to combine terms and concepts. The search terms were as follows:*Concept 1: Dementia* (Dement* OR Alzheimer* OR AD OR DAT OR “Lewy body dementia” OR lewy bod* DLB OR “frontotemporal dementia” OR FTD OR “vascular dementia” OR VD OR “multi infarct dementia”);AND *Concept 2: Social cognition* (Social cogn* OR TOM OR “theory of mind” OR “emo* recogn*” OR empathy OR “emo* perception” OR social function* OR social behavi* OR social activit*);AND *Concept 3: Awareness* (Insight OR conscious* OR aware* OR “self-appraisal” OR “self-reflect” OR anosognosia);AND *Concept 4: Neuroimaging* (Neuroimaging OR imag* OR “magnetic resonance imaging” OR MRI OR “functional magnetic resonance imaging” OR fMRI OR “positron emission tomography” OR PET OR “single-photon emission computed tomography” OR SPECT OR “computed tomography” OR CT OR DTI OR “diffusion tensor imaging” OR DWI OR “diffusion-weighted imaging”).

We hand-searched reference lists of papers identified to be relevant, including previous review articles, to detect additional potentially relevant studies.

### 2.2. Identification and Selection of Studies

Studies were eligible for inclusion if they used structural or functional neuroimaging of any modality to identify neural correlates of awareness of social cognition and function in people with dementia. Only quantitative studies in humans were included and studies had to be peer-reviewed primary research rather than review articles. Dementia diagnoses were required to be made using validated diagnostic criteria. No restriction was made regarding the type of measurement used to assess awareness in social cognition and function, as there is no gold standard tool available [10]. Studies were required to be in the English language, but no restrictions were made regarding year of publication.

The process of selecting and further analysing data from eligible studies was conducted and documented according to the PRISMA specifications for systematic reviews [40] (Appendix A). After screening for relevance, full texts of potentially eligible papers were retrieved and assessed against inclusion criteria by AH and, to ensure a consistent screening procedure at this stage, a randomly selected 10% sample was screened by AS with agreement between reviewers for 41/42 (97.6%) of papers. AH did not exclude any relevant papers so proceeded to screen all remaining articles and any uncertainties were discussed with the research team. Data from studies meeting inclusion criteria were then extracted. A pre-piloted form based on PICOS components was used for data extraction to gather information about the study authors, year of publication, study design, number of participants, mean age, type of dementia diagnosis, diagnostic criteria for dementia, social cognition assessment method, awareness assessment method, neuroimaging methodology, threshold (whole brain/region of interest), results, e.g., x, y, z coordinates (in Montreal Neurological Institute (MNI) or Talairach space), Brodmann’s area, hemisphere, region, and conclusion. The data extraction was carried out independently by two researchers (AH and JH). If disagreements between the two researchers occurred, consensus was reached through discussion and the involvement of a third independent researcher (AS).

All studies included were assessed for quality and risk of bias. A modified version of the Effective Public Health Practice Project (EPHPP) quality assessment tool was used [41,42]. Studies were assessed for: potential for selection bias, appropriateness and rigor of study design, validity of awareness of social cognition and function task, neuroimaging methodology, and appropriateness of statistical analysis used. Each of these subsections were given a subscale quality rating (“high”, “moderate” or “low”), based on which an overall quality rating was calculated (high = no low ratings in any subsection, moderate = one low rating, or low = more than two low ratings) (Appendix A).

### 2.3. Protocol and Registration

A systematic review protocol was registered prospectively with PROSPERO (https://www.crd.york.ac.uk/prospero/display_record.php?RecordID=200503, accessed on 22 August 2022).

## 3. Results

The electronic search across the five databases identified 9679 records and one study was detected by hand-searching reference lists. After duplicates were removed, a total of 9426 titles and abstracts were screened for eligibility, of which 9006 studies were excluded for irrelevance to the research question and 420 papers were identified as potentially relevant. Full text papers were retrieved and assessed for the fulfilment of inclusion criteria; 412 papers were excluded (see PRISMA flowchart in Figure 1). A total of eight papers met the eligibility criteria and were included. The inter-rater reliability for the abstract and full-text screening process was high, with 97.6% agreement, k = 0.9.

### 3.1. Study Characteristics

Six of the eight included studies used structural MRI [43,44,45,46,47,48], one study used PET imaging [49], and another used a combination of MRI and PET [50]. Six studies reported x, y, z coordinates in MNI space, [43,44,46,47,48,49]. Five of these studies directly reported associations between impaired awareness of social cognition and function and structural/functional abnormalities [43,46,47,48,49]. Three studies only allowed inferences to be made based on between-group comparisons of atrophy/hypometabolism and measures of awareness of social cognition and function [44,45,50]. The most common method used to assess awareness of social cognition and function was the patient–informant discrepancy score (n = 7) of social cognition measures [43,44,45,47,48,49], followed by one informant-rated assessment tool. Six studies included participants with multiple subtypes of dementia (e.g., bvFTD, SD, AD, and others), one study only included people with FTD [49], and one study only included people with semantic dementia (SD) [50] (see Table 1). According to our quality rating, two studies were rated with low quality [45,49], two were rated as moderate [44,50], and four were identified as high quality [43,46,47,48].

Four studies were rated as high quality as they did not receive weak ratings in any of the quality domains. Two were rated as moderate, due to one weak rating in the criteria for the neuroimaging methodology [44,50]. Two studies were rated as lower quality, one due to weak ratings in the quality rating criteria for selection, study design, neuroimaging methodology, and analysis [45], and the other for weak ratings in the criteria for selection, study design, and measurements used to assess insight into social cognition [49].

### 3.2. Neural Correlates Identified via Structural Neuroimaging Studies

Four high-quality studies used MRI to assess the neuroanatomical basis of awareness of social cognition and function across FTD subtypes and AD, [43,46,47,48]. All of these used different questionnaires, measuring awareness in various subdomains of social cognition and function. The social cognitive and functional subdomains covered in these questionnaires overlapped, making it difficult to separate them with certainty. Nevertheless, the questionnaires assessed either emotional processing abilities (e.g., empathy, emotion regulation) or social function domains (e.g., social interpersonal functioning, socially appropriate behaviour or self-presentation, meaning the ability to modify one’s behaviour according to the social circumstances) [46,51] (see Table 2). Three studies used patient–informant discrepancy scores [43,47,48], and one study used an informant-rated score to assess lack of awareness in these aspects of social cognition and function [46].

Atrophy in the frontal lobe, in particular, in the frontal pole bilaterally, was found to be associated with impaired awareness of deficits in *emotional processing* abilities in one high-quality study [43]. Impaired awareness of *emotional regulation* abilities correlated with atrophy in the right superior frontal gyrus, the orbital part of the right inferior frontal gyrus and the anterior part of the left insula in another high-quality study [47]. A further high-quality study [48] identified atrophy in the right temporal gyrus, especially the inferior temporal gyrus and the left fusiform gyrus to be associated with impaired awareness of *empathic* abilities. The former high-quality study [43] also found atrophy in the left amygdala and hippocampus to be associated with impaired awareness of deficits of empathy.

Deficits in awareness of *social function* were found to be correlated with left frontal lobe atrophy, particularly in the left orbitofrontal cortex [43], orbitofrontal gyri, and gyrus rectus [46]. In the temporal lobe, atrophy in the right temporal pole [46] and temporal gyri (i.e., left–middle temporal gyrus, fusiform gyrus) [43,46,47] was found to be associated with impaired awareness of social function abilities. Additionally, atrophy in the parahippocampal gyri bilaterally [43,46], right amygdala, bilateral insulae, right occipital cortex [43], and right putamen [47] was associated with an impaired awareness of socially appropriate interpersonal functioning.

In summary, there is evidence from high-quality studies that the neural correlates of awareness of emotional processing abilities and social function overlap. Atrophy in the frontal gyri (i.e., superior orbitofrontal gyrus, inferior orbitofrontal gyrus), temporal gyri (i.e., middle temporal gyrus, inferior temporal gyrus), fusiform gyrus, amygdala, and insula were significantly associated with impaired awareness of both emotional processing and social function in several studies [43,46,47,48].

### 3.3. Neural Correlates Identified via Functional Neuroimaging Studies

Only one study, rated as lower quality [49], included functional neuroimaging, using PET imaging to assess an awareness of deficits in social cognition and function in frontal variant FTD (fvFTD) patients. Awareness of deficits in social behaviour and personality were measured by an informant–patient discrepancy score of a behaviour prediction questionnaire and a personality assessment.

In this study [49], lack of awareness of deficits in social behaviour was associated with reduced metabolic activity in the left inferior and bilateral superior temporal poles. Although deficits in awareness of personality changes were seen consistently in the patients with fvFTD, no significant correlation was found for awareness scores relating to personality traits or metabolic activity in any brain regions.

### 3.4. Inferences of Neural Correlates Based on Group-Wise Comparison across Structural and Functional Studies

In three studies, two of which were moderate [44,50] and one was of low quality [45], inferences of neural correlates of awareness of social cognition and function were made based on group-wise comparisons of regional atrophy or metabolism and scores on measures of awareness of social cognition and function between dementia subtypes and healthy controls.

One moderate quality study [50] used MRI and PET to assess neural correlates of awareness of ToM abilities in an SD group. SD patients demonstrated an impaired awareness of deficits in cognitive ToM domains (i.e., inability to infer intentions or beliefs of someone involved in social scenarios), which was associated with atrophy in the left temporal lobe (including temporal neocortex, hippocampus, fusiform gyrus, insula, caudate, and pallidum nucleus).

Another moderate-quality study [44] used MRI to assess atrophy associated with an impaired awareness of socio-emotional skills (i.e., recognition of emotions, empathy, social conformity, antisocial behaviour and sociability) in patients with behavioural variant FTD (bvFTD), SD and AD. Lack of awareness of deficits in socio-emotional skills was particularly evident in patients with bvFTD, followed by SD. BvFTD patients were noted to have significant atrophy in the right frontal pole, the right central opercular cortex, the right inferior frontal gyrus, the left medial temporal regions (e.g., left temporal fusiform cortex, temporal pole, slight left inferior temporal gyrus), and SD patients showed atrophy in regions including the temporal pole and fusiform cortex.

The lower quality study [45] presented four cases (SD, right temporal lobe variant FTD (rtlvFTD), mild AD, and moderate AD) and reported measures of awareness of social behaviour and neuroimaging findings. Only the rtlvFTD patient demonstrated a profound lack of awareness of changes in their own personality and interpersonal behaviour, and their MRI showed atrophy in the right, and to a lesser extent, the left temporal lobes, possibly indicating brain regions correlated with a loss of awareness of these domains.

Across all three studies, FTD patients had the most significant impairment of awareness of deficits in social cognition and function. This lack of awareness of deficits was assumed to be correlated with the patients´ atrophy in temporal regions (temporal pole, temporal gyri, fusiform gyrus, hippocampal regions) and the insula [44,45,50]. Although all three studies reported similar findings that atrophy in these areas was related to an impaired awareness of social cognition and function, the moderate-quality studies, [44,50] described abnormalities in the left temporal regions (i.e., left temporal poles, temporal gyrus, hippocampal regions, and fusiform gyrus) to be the most relevant.

### 3.5. Comparison across Methodologies

In summary, the brain regions classed as important for awareness of social cognition and function that were identified in more than two studies included the frontal gyrus (i.e., orbitofrontal gyrus) [44,46,47], temporal lobe regions [43,44,45,46,48,49,50], including the temporal pole [44,46,49,50], the temporal gyri (i.e., middle temporal gyrus and inferior temporal gyrus) [44,46,48], the fusiform gyrus [44,47,48,50], and the insula [43,46,47,50]. Brain regions reported by two studies included the frontal pole [43,44], hippocampus, amygdala [43,50], and parahippocampal gyrus [43,46] (schematic representation in Figure 2).

It should be noted that, although these studies all reported these brain regions to be relevant, there was some discrepancy regarding the lateralization of atrophy in some regions. For example, Sollberger et al. [48] identified atrophy in the left fusiform gyrus to be more significant, whilst Shany-ur et al. [47] identified atrophy in the right fusiform gyrus to be more associated with an impaired awareness of social cognition and function (Table 2).

## 4. Discussion

The main findings of this systematic review are that the neural correlates associated with a lack of awareness of social cognition and social function include the frontal pole, orbitofrontal cortex, temporal pole, temporal gyri (middle and inferior temporal gyrus), fusiform gyrus, insula, amygdala, hippocampus, and parahippocampal gyrus. Higher levels of atrophy or abnormal activity in these areas were associated with a more severe lack of awareness of deficits across social cognitive domains and function.

These brain regions were reported in more than one study, using a range of methodologies, structural or functional imaging, or inferences from group comparisons, strengthening the conclusions. Our findings were supported by high-quality studies, and no large differences between the results of lower- and higher-quality studies were found. However, the small number of studies, the heterogeneity of methods used to assess awareness and social cognition, differing neuroimaging methodologies and patient groups, limit confidence in the results. Although the results must be interpreted cautiously, our study confirms and builds upon previous findings, emphasizes the need for further research in this field, and guides future investigations.

First, we identified that the neural correlates of impaired awareness of different social cognition domains and social function overlapped. This may indicate that a specific neural substrate is involved in awareness, or indicate overlap between assessments of different social cognition and function subdomains. In addition, the identified brain regions partially overlapped with the neural correlates of anosognosia and impaired awareness for other cognitive functions in AD (e.g., awareness of memory function) identified by Hallam et al. [38]. Specifically, abnormalities in the superior frontal gyrus, inferior frontal gyrus, orbitofrontal cortex, medial temporal lobe (MTL), and insula were identified as relevant both for impaired awareness of social cognitive and social function deficits in this review and for anosognosia and impaired awareness of cognitive deficits in Hallam et al. [38]. This overlap may indicate that these specific brain regions are important for awareness across cognitive domains or suggest that awareness is a fundamental component of social cognition which cannot be separated from, for example, the ability to recognize emotions in others. Shared findings may also reflect the overlapping nature of assessments of social cognition and awareness of social cognition and function and, partly due to this overlap, our study included two studies which were also in Hallam et al. [38]. However, we also identified atrophy in frontal and temporal poles, fusiform gyrus, hippocampus and parahippocampal gyrus to be more specifically associated with an impaired awareness of social cognition and function, potentially indicating that these areas are linked to domain-specific awareness in dementia, as has been suggested by other groups [26,33].

With respect to the question of whether awareness of social cognition and function can be dissociated from underlying performance of social cognitive processes, several brain regions identified as associated with awareness of social cognition and function overlapped with brain regions important for social cognition, including the orbitofrontal regions, insula and temporal lobe regions, amygdala, and fusiform gyrus [9,35,36,52]. Sollberger et al. [48] have previously reported a significant overlap between the neural correlates of overestimating one’s own empathic abilities and empathy itself. Moreover, Christidi et al. (2018) recently reviewed brain areas associated with deficits in social cognition in neurodegenerative diseases describing, for example, fusiform gyrus damage in primary progressive aphasia which may impair facial recognition and thereby impair social behaviour, and anterior cingulate cortex disruption in AD and FTD which impairs empathic responses to emotional stimuli. There was substantial overlap between the brain areas identified in that review to be associated with deficits in social cognition and the brain areas we identified to be important in deficits in awareness of social cognition and function. We additionally identified hippocampal and parahippocampal atrophy to be associated with an impaired awareness of social cognition and function, but this was not found to be important for studies of social cognition that did not specifically assess awareness, which may indicate a particular metacognitive role for these regions [52].

The OFC has previously been identified to be associated with awareness of social cognition and function as well as awareness in other cognitive domains [38], and is also thought to play a significant role in social cognitive function, per se. For example, activation studies have demonstrated that the OFC is important in several aspects of emotion (i.e., emotion identification, emotion-related learning, social and emotional behaviour, and subjective emotional/affective state) and may play an important role in controlling affective reactions to salient emotional stimuli, such as facial expressions [53,54,55]. Similarly, lesions in the anterior insula and temporal pole regions occurring after acute stroke were associated with impairments in affective empathy [56], supporting the role of these regions in social cognitive processes. In other lesion studies, it has been shown that lesions in the amygdala, for example, caused by encephalitis or neurosurgical treatments for epilepsy, are associated with changes in social cognition domains, in particular, changes in the ability to process social and emotional information and facial emotion expression recognition, so may impact social behaviour [57]. Whilst there is, therefore, evidence for common neural correlates of awareness of social cognition and function, underlying social cognitive processes require further neuroimaging studies using focused methods of assessing awareness of subdomains of social cognition and function to clarify whether awareness can be dissociated from social cognition, per se.

Several brain regions identified in the current review are associated with well-characterized brain networks, such as the default mode network (DMN). The DMN is reported to be involved in social interactions, moral judgement, inferring thoughts and perspectives of other people, and self-referencing [58,59,60]. In addition, impaired connectivity in the DMN is associated with impaired awareness in AD [61]. Brain regions associated with the DMN, including medial temporal lobe regions, the hippocampus, and the parahippocampus [62], were identified to be associated with an impaired awareness of social cognition and function in our review.

Our results are also in keeping with some theoretical models of awareness, such as the cognitive awareness model (CAM), which provides a neurocognitive explanation for deficits in awareness based on a range of potential impairments in memory and metacognitive systems, preventing the storage or updating of relevant information about the self [63]. Awareness may, therefore, be impaired due to abnormalities in different brain regions or networks, supporting these mnemonic or metacognitive systems. For example, important regions involved in the CAM model are frontal and medial temporal structures, which we also found to be important in the loss of awareness of social cognition and function. This may reflect some common mechanisms of impaired storage or updating of knowledge of the self across general cognitive and social cognition domains, whereas the results of the present study found no clear evidence that an awareness of social cognition and function is distinct from its underlying social cognitive domains; the heterogeneity of neurodegeneration in the brain regions identified may be related to the differing patterns of loss of awareness of social cognition and function within and between various dementia types [61,63].

Due to some studies not reporting the neural correlates associated with an awareness of social cognition and function separately for each dementia type, comparison across dementia diagnosis is limited. However, the findings of the present study confirm previous results, in which FTD patients were found to have the most profound lack of awareness of social cognition and function compared to other dementia types or controls.

### 4.1. Limitations

The study has limitations. Whilst the quality of the small number of included studies was good, with four studies being identified as high quality, the studies included had relatively small sample sizes, with particularly small dementia subgroups, which might have underpowered specific findings.

In addition, there was variability in the measurement tools used for the assessment of awareness of social cognition and function. As can be seen from Table 2, in most studies, discrepancy scores between patient and informant were used to measure awareness of social cognition and function, and every study used a different questionnaire investigating specific concepts of social cognition. It was, therefore, difficult to compare study findings and draw definitive conclusions about which areas of the brain are relevant to the awareness of specific social cognitive subdomains and social function. This is further limited by the fact that social cognitive subdomains may not be distinct and, therefore, the attempt to separate social cognitive domains may be challenging. In addition, it is not established if social cognition may intrinsically involve awareness and it remains unclear whether these concepts can be separated. Moreover, the lack of an objective assessment of awareness may result in potential bias and does not allow for the control of the severity of deficits in social cognition. Overall, the heterogeneity of the tools used is a significant limitation of the existing literature.

The absence of a single established questionnaire or gold-standard tool for awareness of social cognition and function, and the complexity of the concept of awareness [33,64] precludes meta-analysis and points to a need for future research to identify and consistently use gold-standard assessments of awareness. The limited number of studies available and their lack of specific neuroanatomical details, as well as the inconsistent use of terminologies across the papers, further precluded meta-analysis. In this review, only one study used functional neuroimaging to detect specific neural correlates [49]. Moreover, inferences made based on group-wise comparisons are limited; for example, patients classified into diagnostic categories by imaging findings, e.g., FTD patients diagnostically classified by frontal atrophy, then had phenotypic characteristics linked to those findings. This review is further limited by the fact that mainly FTD patients were assessed, as deficits in social cognition and loss of awareness are major symptoms in FTD, occurring earlier than in other dementia types [25]. We had planned to examine how neural correlates of impaired awareness of social cognition varied across different dementia subtypes but the small number of studies of non-FTD dementias precluded this. Our findings about the lateralization of neural correlates of impaired awareness of social cognition and function are less certain, as our findings are limited to the results presented in included studies and several studies did not report the handedness of participants.

Generalizability to other dementia subtypes is unclear and a future research focus should be investigating neural correlates of awareness of social cognition and function in larger cohorts of other dementia types.

### 4.2. Future Research and Implications

This review is the first, to our knowledge, that focuses on identifying neural correlates of awareness of social cognition and function, and thus contributes to a better understanding of the deficits in awareness of social cognition and function in dementia and highlights gaps in the neurocognitive literature. Given the limited number and limitations of the existing studies, our conclusions are cautious and further research is required into the structural and functional abnormalities underlying impairments of awareness of social cognition and function in different types of dementia. More high-quality studies are needed and future studies may focus on using structural and, in particular, functional neuroimaging techniques, and may consider a longitudinal assessment to identify the changes in neural correlates with dementia progression. Studies should establish and use gold-standard tasks assessing the awareness of subdomains of social cognition and function in conjunction with functional methodologies to elucidate these relationships. More research is needed to fully understand the concept of awareness in social cognition and function, as well as the distinction between subdomains of social cognition (i.e., social cognition and function). In addition, a consistent use of more precise brain region localizations using standardized terminologies across research studies would be important, allowing for better comparisons of findings and enabling meta-analysis, for example, by using the automated anatomical labelling atlas (AAL3) [65]. Identifying the brain regions associated with an awareness of social deficits can contribute to understanding the complex construct of awareness and its differentiation from social cognition.

Improving the characterization of how neurodegeneration in dementia leads to social cognition deficits, compounded by a lack of awareness, is also relevant for clinical practice. A lack of awareness of cognitive and functional deficits due to dementia has negative effects on outcomes for the person with dementia and their wider family, which may relate to worse neuropsychiatric symptoms, difficulty managing medication, impaired safety management and self-care, and higher carer stress [66,67]. An improved understanding of the nature and causes of this impaired awareness in dementia has potential to lead to better care [10] and potentially reduced care costs [68]. It may be beneficial to be able to describe to patients and their families that the impaired awareness they experience is a direct result of dementia pathology so that they can recognise this as a symptom of the condition and find adaptive strategies, potentially enhancing quality of life, reducing carer burden, and improving care [10]. Future development and testing of interventions for people with dementia who lack awareness of their impaired social cognition and function are required to improve outcomes for people with dementia.

## Figures and Tables

**Figure 1 brainsci-12-01136-f001:**
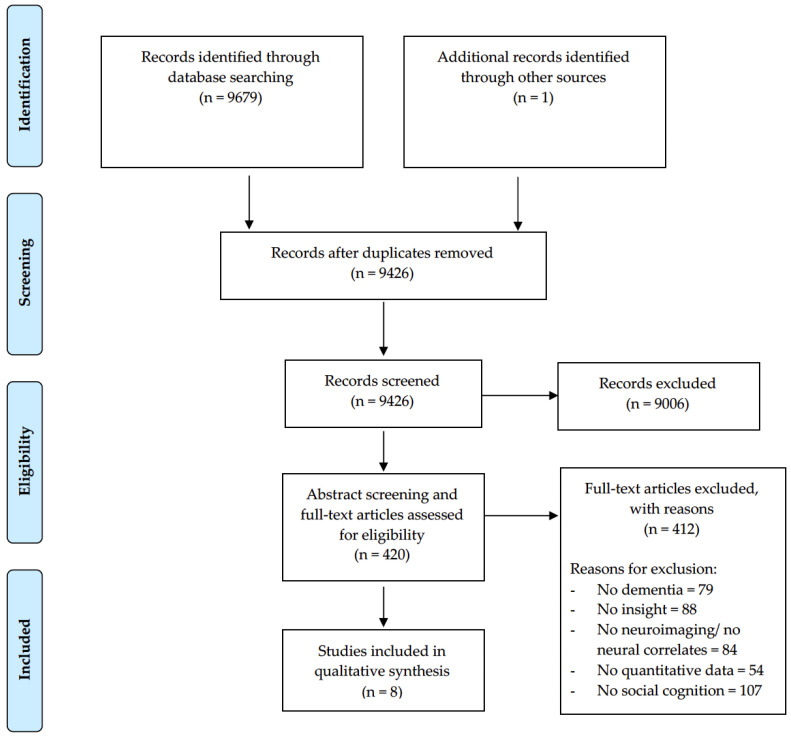
PRISMA flowchart for study selection for systematic review.

**Figure 2 brainsci-12-01136-f002:**
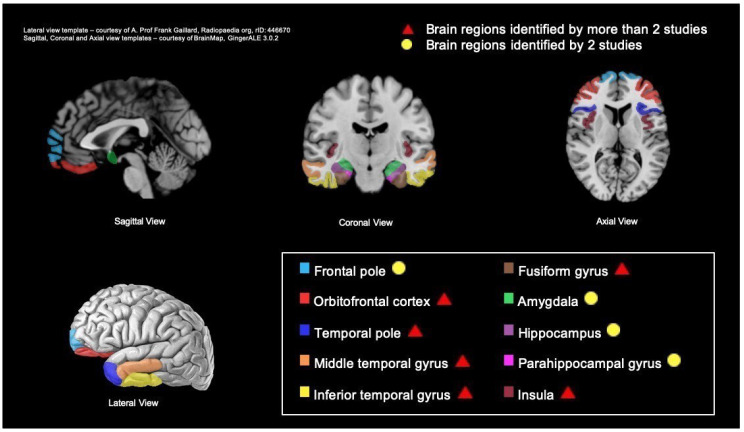
Schematic representation of brain regions where atrophy or hypometabolism is associated with impaired awareness of social cognition and function in several dementia types (adapted from [38]).

**Table 1 brainsci-12-01136-t001:** Study characteristics.

Authors	Number of Patients and Type of Diagnosis	Mean Age of Patient (Standard Deviation (SD))	Gender (M/F)	Mean MMSE Dementia Severity (SD)	Type of Scan	Awareness Measure		Quality Assessment (Score)
						Type	Measure	
Duval et al. [50]	15 SD	64.27 (6.5)	6/9	n/a	MRI, PET	Patient–carer discrepancy score	Attribution of Intention Test, False Belief Test, Reading the Mind in the Eyes Test	Moderate (7)
Hornberger et al. [43]	81				MRI	Patient–carer discrepancy score	28-item Insight Questionnaire	High (9)
	24 bvFTD	63.13 (10.8)	18/6	25.55 (3.7)				
	18 SD	64.11 (9.2)	13/5	24.06 (2.5)				
	13 PNFA	63.64 (8.4)	9/4	26.09 (2.9)				
	15 AD	64.27 (9.1)	12/3	22.07 (7.6)				
	11 LPA	63.64 (8.4)	4/7	22.18 (3.5)				
Hutchings et al. [44]	41			n/a	MRI	Patient–carer discrepancy score	Socio-emotional Questionnaire (SEQ)	Moderate (8)
	16 bvFTD	64.05 (10.3)	12/4					
	15 SD	64.05 (5.9)	8/7					
	10 AD	66.25 (9.1)	4/6					
Mychack et al. [45]	4				MRI	Patient–carer discrepancy score	Interpersonal Adjectives Scale (IAS), Interpersonal Measure of Psychopathy (IMP)	Low (2.5)
	1 SD	68	M	23				
	1 rtlvFTD	52	M	30				
	1 mild AD	70	F	27				
	1 moderate AD	84	F	14				
Parthimos et al. [46]	77 39 AD 38 FTD (13bvFTD, 11 svPPA, 4 nfvPPA, 6 CBS, 4PSP)	n/a	26/51	n/a	MRI	Informant-rated score	Revised Self-monitoring Scale (RSMS)	High (10)
Ruby et al. [49]	21 FTD	64 (9)	13/8	n/a	PET	Patient–carer discrepancy score	Behaviour prediction Questionnaire, Personality assessment Questionnaire	Low (7)
Shany-ur et al. [47]	78				MRI	Patient–carer discrepancy score	Patient Competency Rating Scale (PCRS)	High (12)
	35 AD	64.8 (8.6)	17/18	23.4 (3.8)				
	21 bvFTD	59.7 (7.2)	12/9	24.9 (3.6)				
	7 rtvFTD	61.9 (6.9)	3/4	27 (1.2)				
	8 svPPA	57.9 (6.6)	4/4	21.8 (8.2)				
	7 nfvPPA	66 (9.2)	4/3	24.9 (6)				
Sollberger et al. [48]	83				MRI	Patient–carer discrepancy score	Interpersonal Reactivity index (IRI)	High (10)
	28 bvFTD	62.4 (8.2)	21/7	25.9 (4.7)				
	16 svPPA	61.8 (6.7)	10/6	25.3 (5.5)				
	4 nfvPPA	62.0 (9.4)	2/2	27.0 (3.6)				
	23 AD	63.3 (10.3)	15/8	19.9 (6.3)				
	12 CBS	66.8 (9.2)	4/8	22.6 (7.1)				

Note. MMSE = Mini Mental State Examination (max point 30; mild = 21–26; moderate = 10–20; moderately severe = 10–14; severe = less than 10); SD = semantic dementia; bvFTD = behavioural variant frontotemporal dementia; PNFA = progressive nonfluent aphasia; AD = Alzheimer´s disease; LPA = logopenic progressive aphasia; rtlvFTD = right temporal lobe variant of frontotemporal dementia; svPPA = semantic variant primary progressive aphasia; nvPPA = nonfluent variant of primary progressive aphasia; CBS = corticobasal syndrome; PSP = progressive supranuclear palsy; rtvFTD = right temporal variant frontotemporal dementia.

**Table 2 brainsci-12-01136-t002:** Neural correlates of awareness of social cognition and function subdomains identified by structural and functional neuroimaging.

Study	Domain	Assessment Method	Type of Scan	Coordinates (MNI)			Regions
				x	y	z	
**Structural studies**	**Awareness of emotion processing abilities**						
Hornberger et al. [43]		28-item Insight Questionnaire *Subscale* • Emotion (Empathy)	MRI	−20	66	0	*L Frontal pole*
				22	64	−6	R Frontal pole
				−24	−4	−10	L Amygdala
				−34	−10	−16	L Hippocampus
Shany-ur et al. [47]		Patient Competency Rating Scale (PCRS) *Subscale* • Emotion regulation (e.g., accepting criticism from others)	MRI	38	24	−10	R Inferior frontal gyrus, orbital part
				40	20	−2	R Insula, anterior
				18	66	22	R Superior frontal gyrus
				−30	26	−26	L Insula, anterior
				−25	28	−16	L Inferior frontal gyrus, orbital part
Sollberger et al. [48]		Interpersonal reactivity index (IRI)	MRI	60	6	−34	R Inferior temporal gyrus
				−30	−36	−14	L Fusiform gyrus
	**Awareness of social function**						
Hornberger et al. [43]		28-item Insight Questionnaire *Subscale* • Socially appropriate interaction	MRI	−28	−8	−36	L Parahippocampal Gyrus
				−16	10	−18	L Orbitofrontal Cortex
				−44	−58	−6	L Temporal Gyrus
				−46	−12	2	L Insular Cortex
				64	−12	−34	R Temporal Gyrus
				34	−2	−18	R Amygdala
				28	−58	34	R Occipital Cortex
Parthimos et al. [46]		Revised Self-monitoring Scale (RSMS) *Subscale* • Self-presentation	MRI	27	15	−27	
							R Insula
							R Superior temporal pole
							R Orbitofrontal gyrus
							R Parahippocampal gyrus
							R Rectus
							R Olfactory
				−16.5	9	−16.5	
							L Insula
							L Olfactory
							L Rectus
							L Orbitofrontal gyrus
				−63	−19.5	−12	L Middle temporal gyrus
Shany-ur et al. [47]		PCRS *Subscale* • Social interpersonal functioning (e.g., participating in group activities)	MRI	24	10	−4	R Putamen
				28	−12	−40	R Fusiform gyrus
**Functional study**	**Awareness of social function**						
Ruby et al. [49]		Behaviour Prediction Questionnaire • Anosognosia for behavioural change (S1-R3)	PET	−40	12	−20	L Superior part of the temporal pole
				−58	−10	−30	L Inferior part of the temporal pole
				56	18	−14	R Superior part of the temporal pole

Note: MRI = magnetic resonance imaging; PET = positron emission tomography; MNI = Montreal Neurological Institute and Hospital coordinate system.

## Data Availability

All results presented in this systematic review are included in the Appendix A.

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
