# Peer review of "What Are the Neural Correlates of Impaired Awareness of Social Cognition and Function in Dementia? A Systematic Review"

_brainsci, 2022, doi:10.3390/brainsci12091136_

Round 1

Reviewer 1 Report

The authors present a well performed systematic review on neural correlates of impaired awareness of social cognition and function in dementia. The review is well written.

In the results section the authors sum up neural correlates, partially pointing out if correlates are found left or right, partially lateralization is not mentioned. In those cases I would suggest to use “bilateral” to point out more clearly, that not lateralization is found, which is especially relevant in cases, where opposed lateralization is seen. This is shorty mentioned in the results section, but should be discussed further on in the discussion section.

Author Response

Reviewer #1: The authors present a well performed systematic review on neural correlates of impaired awareness of social cognition and function in dementia. The review is well written.

Response: Thank you very much for your positive comments and helpful suggestions to improve our manuscript. We have addressed your comments below (in bold type) and detailed the changes we have made to the manuscript as a result (in italics).

Point 1: In the results section the authors sum up neural correlates, partially pointing out if correlates are found left or right, partially lateralization is not mentioned. In those cases I would suggest to use “bilateral” to point out more clearly, that not lateralization is found, which is especially relevant in cases, where opposed lateralization is seen. This is shorty mentioned in the results section, but should be discussed further on in the discussion section.

Response 1: Thank you for this suggestion. This information was previously presented in table 2 and some of the detail was omitted from the text, but we agree it is helpful to provide this for the reader so have now added detail about the lateralisation of lesions in several throughout the results section and have described “bilateral” neural correlates where this is the case.

Our review summarises the results presented from the included studies so our conclusions about lateralisation, including partial lateralisation, are limited by the reporting in those studies. This meant that it is difficult to be certain to what extent impaired awareness of social cognition and function is driven by lateralised lesions, and studies frequently did not report the handedness of participants. We have added this limitation to the manuscript:

“Our findings about the lateralization of neural correlates of impaired awareness of social cognition and function are less certain as our findings are limited to the results presented in included studies and several studies did not report the handedness of participants.”

Reviewer 2 Report

Dear Authors,

It was a pleasure reviewing your work. I found it scientifically sound, well written and organized.

Below are some minor remarks related to the article.

1) I suggest to at least mention other types of dementia alongside FTD and AD in the introduction (lines 31-34).

2) Did the search also include 'VaD' term for vascular dementia? I did not find it in the listed search. 

3) What was the argument for not including meta-analyses in your literature search? Some additional studies could be handpicked this way. 

If meta-analyses were also considered, please state it more clearly in the methods section.

4) Line 163: What was the agreement score after random screening of 10% of the articles? I suggest to include this piece of information.

5) My final and most important remark is: 

With the thorough, organized and logically sound content of the article, the part related to clinical implications seems somewhat insufficient. It would be valuable to discuss or at least consider some other implications of these findings. I recommend to extend the information provided at the end of conclusions.

Author Response

Reviewer #2: Dear Authors,

It was a pleasure reviewing your work. I found it scientifically sound, well written and organized.

Below are some minor remarks related to the article.

Response: Thank you very much for your positive comments and helpful suggestions to improve our manuscript. We have addressed your comments point by point below (in bold type) and detailed the changes we have made to the manuscript as a result (in italics).

Point 1: I suggest to at least mention other types of dementia alongside FTD and AD in the introduction (lines 31-34).

Response 1: Thank you, we expanded this paragraph:

“The Diagnostic and Statistical Manual of Mental Disorders (DSM-5) criteria for dementia, cites social cognition as one of the core domains of impaired neurocognitive function [4] and, in frontotemporal dementia (FTD), early decline in social conduct is a core diagnostic feature [5] , with changes in social cognitive abilities commonly reported in behavioral-variant FTD (bvFTD) and semantic dementia (SD) [6,7]. Impairments of social cognitive processes and social function are also common in Alzheimer's disease (AD), though are usually not as severe and occur later in the disease course than in FTD [2,9–11] and social cognitive decline also occurs in dementia with lewy bodies (DLB) [8].”

Point 2: Did the search also include 'VaD' term for vascular dementia? I did not find it in the listed search. 

Response 2:  We did not use the abbreviation “VaD” in our search terms but used other terms commonly used in the literature - “vascular dementia”, “ VD”, “multi infarct dementia” – to search for vascular dementia.

Point 3: What was the argument for not including meta-analyses in your literature search? Some additional studies could be handpicked this way. If meta-analyses were also considered, please state it more clearly in the methods section.

Response 3: Our systematic review sought quantitative primary research papers so we did not include any meta-analyses in our review – although we do not think that there are any existing meta-analyses of studies exmining this research question which was our motivation to conduct our own review. We hand searched the reference lists from relevant papers including previous review articles seeking any additional potentially relevant studies for inclusion in our review and this is stated in our methods section:.

“We hand-searched reference lists of papers identified to be relevant, including porevious review articles to detect additional potentially relevant studies.”

Point 4: Line 163: What was the agreement score after random screening of 10% of the articles? I suggest to include this piece of information.

Response 4: Thank you very much for pointing this out, we are sorry that this was not clearly stated. There was agreement in 98% of screening and the main reviewer (AH) did not exclude any relevant articles so we judged that it was appropriate for AH to proceed to screen all remaining articles. We have added this:

“After screening for relevance, full texts of potentially eligible papers were retrieved and assessed against inclusion criteria by AH and, to ensure a consistent screening procedure at this stage, a randomly selected 10% sample was screened by AS with agreement between reviewers for 41/42 (98%) of papers. AH did not exclude any relevant papers so proceeded to screen all remaining articles and any uncertainties were discussed with the research team.”

Point 5: My final and most important remark is: 

With the thorough, organized and logically sound content of the article, the part related to clinical implications seems somewhat insufficient. It would be valuable to discuss or at least consider some other implications of these findings. I recommend to extend the information provided at the end of conclusions.

Response 5: Thank you very much for this helpful comment. We agree that an extension of the paragaph on clinical implications is only beneficial. Thus, we expanded this:

“Improving the characterization of how neurodegeneration in dementia leads to social cognition deficits, compounded by a lack of awareness is also relevant for clinical practice. Lack of awareness of cognitive and functional deficits due to dementia has negative effects on outcomes for the person with dementia and their wider family which may relate to worse neuropsychiatric symptoms, difficulty managing medication, impaired safety management and self-care, and higher carer stress [67,68]. Improved understanding of the nature and causes of this impaired awareness in dementia has potential to lead to better care [40] and potentially reduced care costs [69]. It may be beneficial to be able to describe to patients and their families that the impaired awareness they experience is a direct result of dementia pathology so that they can recognise this as a symptom of the condition and find adaptive strategies, potentially enhancing quality of life, reducing carer burden, and improving care [40]. Future development and testing of interventions for people with dementia who lack awareness of their impaired social cognition and function are required to improve outcomes for people with dementia.”